# Combination of bortezomib and venetoclax targets the pro-survival function of LMP-1 and EBNA-3C of Epstein-Barr virus in spontaneous lymphoblastoid cell lines

Kam Pui Tam[1,☯], Jia Xie[1,☯], Rex Kwok Him Au-Yeung[2]*, Alan K. S. Chiang[1]*

1 Department of Pediatrics and Adolescent Medicine, School of Clinical Medicine, LKS Faculty of Medicine, The University of Hong Kong, Hong Kong SAR, China, 2 Department of Pathology, School of Clinical Medicine, LKS Faculty of Medicine, The University of Hong Kong, Hong Kong SAR, China

☯ These authors contributed equally to this work.
* rex.auyeung@hku.hk (RKHA); chiangak@hku.hk (AKSC)

**Data Availability Statement:** The data generated in this study are available within the article and its supplementary data files. RNA-seq data was

## Abstract

Epstein-Barr virus (EBV) manipulates the ubiquitin-proteasome system and regulators of Bcl-2 family to enable the persistence of the virus and survival of the host cells through the expression of viral proteins in distinct latency patterns. We postulate that the combination of bortezomib (proteasome inhibitor) and venetoclax (Bcl-2 inhibitor) [bort/venetoclax] will cause synergistic killing of post-transplant lymphoproliferative disorder (PTLD) through targeting the pro-survival function of latent viral proteins such as latent membrane protein-1 (LMP-1) and EBV nuclear antigen-3C (EBNA-3C). Bort/venetoclax could synergistically kill spontaneous lymphoblastoid cell lines (sLCLs) derived from patients with PTLD and EBV-associated hemophagocytic lymphohistiocytosis by inducing DNA damage response, apoptosis and G1-S cell cycle arrest in a ROS-dependent manner. Bortezomib potently induced the expression of Noxa, a pro-apoptotic initiator and when combined with venetoclax, inhibited Mcl-1 and Bcl-2 simultaneously. Bortezomib prevented LMP-1 induced proteasomal degradation of IκBα leading to the suppression of the NF-κB signaling pathway. Bortezomib also rescued Bcl-6 from EBNA-3C mediated proteasomal degradation thus maintaining the repression of cyclin D1 and Bcl-2 causing G1-S arrest and apoptosis. Concurrently, venetoclax inhibited Bcl-2 upregulated by either LMP-1 or EBNA-3C. Bort/venetoclax decreased the expression of phosphorylated p65 and Bcl-2 at serine 70 thereby suppressing the NF-κB signaling pathway and promoting apoptosis, respectively. These data corroborated the marked suppression of the growth of xenograft of sLCL in SCID mice (p<0.001). Taken together, the combination of bortezomib and venetoclax targets the pro-survival function of LMP-1 and EBNA-3C of Epstein-Barr virus in spontaneous lymphoblastoid cell lines.

## Author summary

Epstein-Barr virus (EBV) is an oncogenic virus associated with different cancers and can directly drive the development of a lymphoma type condition in organ transplant patients

deposited to NCBI BioProject database and is available under accession number PRJNA737310.

**Funding:** This research was funded by grants from Health and Medical Research Fund (HMRF) grant #18170462 and grant #20190432; Epstein–Barr virus (EBV) research grant #200004525 of A.K.S. Chiang. The PhD study of Kam Pui Tam has been supported by UPAM Edward & Yolanda Wong Fund and The University of Hong Kong Postgraduate Scholarship. The PhD study of Jia Xie has been supported by The University of Hong Kong Postgraduate Scholarship. The funders had no role in the study design, data collection and analysis, the decision to publish or preparation of the manuscript.

**Competing interests:** The authors have declared that no competing interests exist.

known as post-transplant lymphoproliferative disorder (PTLD) as a result of weakened immune surveillance of EBV. The malignant spectrum of PTLD will require multi-agent chemotherapy regimen which is not well tolerated by the immunocompromised patients. Here, we combined bortezomib (proteasome inhibitor) with venetoclax (Bcl-2 inhibitor) as a novel strategy to target the pro-survival function of key EBV onco-proteins, namely LMP-1 and EBNA-3C. We demonstrated that this chemotherapy-free regimen could be highly effective in killing patient-derived spontaneous lymphoblastoid cell lines which represent cell models of PTLD. The novel drug regimen can avoid the toxic effects of chemotherapy and may have high efficacy and specificity for the treatment of PTLD.

## Introduction

Epstein-Barr virus (EBV), a ubiquitous gamma herpesvirus, latently infects normal B cells and establishes life-long persistence in over 90% of the world population. Though it usually leads to asymptomatic infection in childhood, it is also strongly associated with various lymphoid and epithelial cancers including endemic Burkitt's lymphoma, Hodgkin lymphoma, diffuse large B cell lymphoma, nasal NK/T-cell lymphoma, post-transplant lymphoproliferative disorder, nasopharyngeal carcinoma and a subset of gastric carcinoma [1–3]. EBV establishes different latency patterns by expressing different combinations of viral latent proteins to drive the pathogenesis of these cancers. EBV expresses full spectrum of latent genes, including latent membrane proteins (LMP-1, -2A and -2B), EBV nuclear antigens (EBNA-1, -2, -LP, -3A, -3B and -3C), EBV-encoded small RNAs (EBERs) and BamHI-A rightward transcripts (BARTs) to establish latency III state and contribute to uncontrolled proliferation of B cells in immuno-compromised host. Such latency is found in AIDS-associated B-cell lymphoma, post-transplant lymphoproliferative disorder (PTLD) and lymphoblastoid cell line (LCL) [4,5] which can serve as an *in vitro* model of PTLD.

Pediatric transplant patients who are seronegative to EBV have higher risk of developing PTLD. Rituximab, a chimeric anti-CD20 monoclonal antibody, can be an effective agent alone or in combination with chemotherapy in a proportion of the cases of PTLD. However, there is higher risk of infection and neutropenia associated with treatment regimen containing rituximab [6,7]. Moreover, intensive chemotherapy regimen can cause high toxicity in the immuno-compromised patients [8]. Adoptive immunotherapy employing EBV-specific cytotoxic T cells can successfully target viral antigens and eradicate EBV-positive B cells but the extended turnaround time and requirement of specialized centers limit the widespread application of this therapeutic strategy [9,10]. We are interested in therapeutic strategy that counteracts the survival functions conferred by EBV in EBV-driven lymphoproliferative diseases (EBV-LPDs). Our group previously demonstrated that combination of histone deacetylase and proteasome inhibitors [SAHA/bortezomib] preferentially induced synergistic killing in LCLs and Burkitt's lymphoma (BL) cells of type III or Wp-restricted latency [11]. Such synergism vanished in EBNA-3C knockout BL cells indicating that SAHA/bortezomib might target the survival pathways activated by EBNA-3C [12]. Nevertheless, we could not verify this in LCLs as SAHA/bortezomib induced cell death synergism in spontaneous lymphoblastoid cell line (sLCL) irrespective of the expression of EBNA-3C.

Bortezomib is the first proteasome inhibitor approved by FDA in 2003 and proves to be effective for treatment of numerous types of cancers [13]. Inhibition of NF-κB signaling pathway is the major anti-cancer mechanism of bortezomib [14] and additionally, bortezomib also induces the expression of phorbol-12-myristate-13-acetate-induced protein 1 (PMAIP1), also

known as Noxa, a pro-apoptotic BH3-only protein to inhibit the pro-survival guardian, Mcl-1 and thus activates apoptosis [15,16]. Venetoclax (ABT-199) specifically inhibits Bcl-2 and is approved by FDA in 2016 for treatment of relapsed or refractory chronic lymphocytic leukemia [17]. We propose to employ the combination of proteasome and Bcl-2 inhibitors, bortezomib and venetoclax (bort/venetoclax) to target the pro-survival function of specific EBV proteins in LCLs.

Several EBV latent proteins, LMP-1, EBNA-3C and BHRF-1, confer pro-survival function to EBV-driven B cell lymphoproliferation. LMP-1 is a constitutively active CD40 receptor homolog, transduces signal to promote the proteasomal degradation of inhibitor of nuclear factor kappa B (IκBα) and induces expression of Bcl-2 [18], Bfl-1 [19] and Mcl-1 [20] as well as downregulates Bax [21] to inhibit apoptosis and promote cell survival. EBNA-3C binds to the promoter of Bcl-6 to repress its transcriptional activity [22,23] and mediates proteasomal degradation of Bcl-6. Cyclin D1 and Bcl-2, which are the repression targets of Bcl-6, are activated to facilitate G1-S transition and anti-apoptosis in the LCLs [24]. EBNA-3C also attenuates DNA damage response and G2/M arrest through down-regulation and proteasomal degradation of p53 and H2AX [25–28]. BHRF1 has anti-apoptotic activity similar to Bcl-2 and can directly sequester pro-apoptotic proteins such as Bim, Puma, Bid and Bak [29,30].

In this study, we hypothesized that bort/venetoclax could synergistically kill EBV-positive cells such as LCL by targeting the pro-survival function of LMP-1 and EBNA-3C. In brief, bort/venetoclax could synergistically kill sLCLs derived from patients with PTLD and EBV-associated hemophagocytic lymphohistiocytoisis by inducing DNA damage response, apoptosis and G1-S cell cycle arrest in a ROS-dependent manner. Bortezomib prevented LMP-1-induced proteasomal degradation of IκBα leading to the suppression of the NF-κB signaling pathway and rescued Bcl-6 from EBNA-3C-mediated proteasomal degradation whilst venetoclax inhibited Bcl-2 upregulated by either LMP-1 or EBNA-3C. Taken together, the combination of bortezomib and venetoclax targets the pro-survival function of LMP-1 and EBNA-3C of Epstein-Barr virus in post-transplant lymphoproliferative disorder.

## Results

### Bortezomib/venetoclax induced cell death synergism in spontaneous lymphoblastoid cell lines

Aiming at targeting the pro-survival function of EBNA-3C and other viral proteins such as LMP-1 and BHRF-1 in spontaneous lymphoblastoid cell lines (sLCLs) to induce synergistic cell death, we examined the effects of bortezomib/venetoclax (bort/venetoclax) in sLCLs. The cell viability and dose-dependent cytotoxicity of either bortezomib, venetoclax alone or combination in sLCL 381 and sLCL 421 was determined by MTT assays (Fig 1a). Synergism was observed in all these sLCLs as shown by isobologram analysis and combination indices of less than 1 suggesting that bort/venetoclax might counteract the pro-survival function of the viral proteins in the sLCLs (Fig 1b). The synergism was also evaluated by SynergyFinder, and the synergy score of sLCL381 was 16.455 +- 1.23 and that of sLCL421 was 10.289 +- 1.91, which were greater than 10, supporting that the drug combination has synergistic action (S8 Fig). The combination of 10nM bortezomib and 5μM venetoclax were identified as the optimum concentration because it was the minimum concentration of each drug to achieve approximately 50% cytotoxicity in both sLCL381 and sLCL421. The combination of 10nM bortezomib and 5μM venetoclax was tested on 4 new sLCLs derived from PTLD and EBV-HLH patients, which were sLCL 366, sLCL 397, sLCL 401, and sLCL478. The results showed that the combination treatment of 10nM bortezomib and 5μM venetoclax yielded similar level of viability (around 50%) in the 4 sLCLs as that of sLCL 381 and sLCL 421 (S6 Fig).

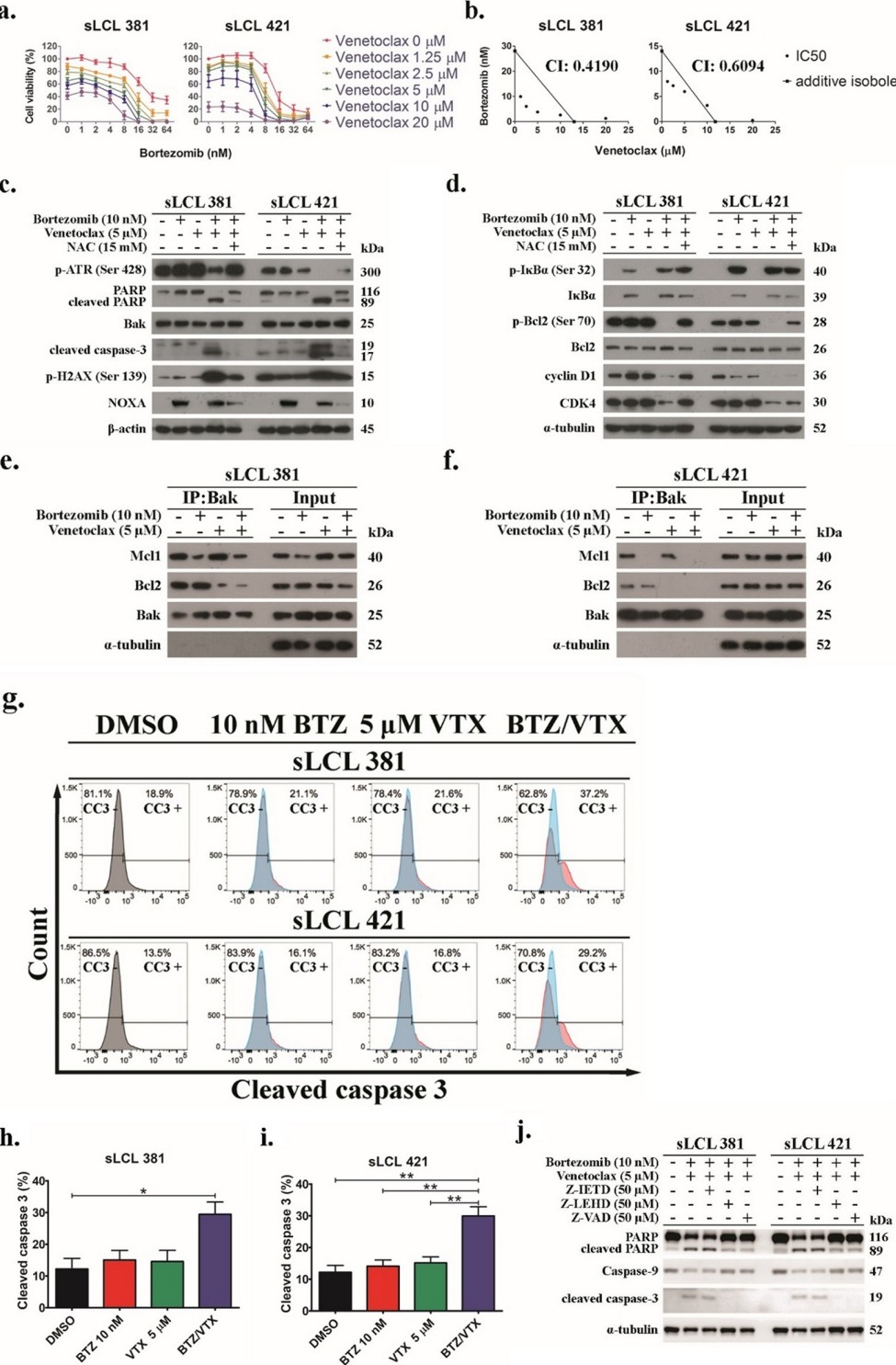

**Fig 1. Bort/venetoclax promote synergistic cell death through inducing expression of apoptotic and DNA damage response markers as well as inhibitor of NF-κB but suppressed regulators for cell cycle progression in sLCLs.** sLCL 381 and sLCL 421 were treated with combination of bortezomib (0, 1, 2, 4, 8, 16, 32, and 64 nM) and venetoclax (0, 1.25, 2.5, 5, 10, 20 μM) for 24 hours and stained by MTT solution. OD570 nm and OD630 nm were measured, and cell viability was plotted. **(a)** Percentages of cell viability in bort/venetoclax treatments were determined. **(b)** Isobolograms and CI were applied for analysis of synergism of bort/venetoclax. **(c-d)** sLCL 381 and sLCL 421 cells were treated with

conditions as described with or without 15 mM N-acetylcysteine (NAC). Proteins were extracted and subjected to western blot analysis with indicated antibodies. **(e-f)** sLCL 381 and sLCL 421 cells were treated with conditions as described. Proteins were extracted and subjected to co-immunoprecipitation with anti-Bak antibody. The expression of Mcl-1, Bcl-2 and Bak were detected by western blot analysis. **(g)** sLCL 381 and sLCL 421 cells were treated with conditions as described. The cells were fixed, incubated with anti-cleaved caspase-3 antibody overnight, and then probed by AF488-conjugated anti-rabbit secondary antibody for cytometric analysis. One representative set of flow cytometric analysis of the cell lines was presented. Percentage of cleaved caspase-3 in **(h)** sLCL 381 and **(i)** sLCL 421 cells upon the treatments was analyzed for statistical significance by using One-way ANOVA Bonferroni post-tests. *$p < 0.05$ was considered as statistically significant (*$p < 0.05$ and **$p < 0.01$). **(j)** sLCL 381 and sLCL 421 cells were either treated with DMSO or bort/venetoclax with or without 50 μM caspase-8 inhibitor (Z-IETD), 50 μM caspase-9 inhibitor (Z-LEHD), or 50 μM pan-caspase inhibitor (Z-VAD). Proteins were extracted and subjected to western blot analysis with indicated antibodies.

RNA-seq was performed to examine the differential gene expression in sLCL 381 and sLCL 421 upon treatment of either DMSO or bort/venetoclax for 24 hours. Pathway enrichment illustrated that bort/venetoclax induced differentially expressed genes related to pathways of EBV infection and cell cycle with the highest enrichment scores (S9 Fig). Of note, the transcription of cell cycle, apoptosis and NF-κB regulators, which is dysregulated by EBNA-3C and LMP-1, was significantly altered upon the treatment of bort/venetoclax (S10 Fig).

Western blot analysis demonstrated that bort/venetoclax induced stronger expression of cleaved caspase-3 (CC3), cleaved poly (ADP-ribose) polymerase (c-PARP) and p-H2AX (Ser139) but the p-ATR (Ser428) was decreased in the sLCLs suggesting that bort/venetoclax potently induced apoptosis and DNA damage response (DDR) but suppressed PARP and the activation of ataxia telangiectasia mutated and Rad3-related (ATR) for DNA repair. Additionally, Noxa was increased upon the treatment of bortezomib (Fig 1c). Bortezomib also prevented the proteasomal degradation of IκBα. The expression of Bak and Bcl-2 remained unchanged whilst the p-Bcl-2 (Ser70) was strongly reduced upon the treatment of bort/venetoclax. Furthermore, the expression of cyclin D1 and cyclin-dependent kinase 4 (CDK4), which are responsible for promoting G1-S cell cycle transition, was diminished only upon the treatment of bort/venetoclax indicating G1 arrest (Fig 1d). Such DDR, apoptosis, and cell cycle arrest were driven by reactive oxygen species (ROS) induced by bort/venetoclax because these responses were diminished when the sLCLs were treated with N-acetylcysteine (NAC), a ROS scavenger (Fig 1c and 1d). Co-immunoprecipitation demonstrated that bortezomib reduced the binding of Mcl-1 to Bak whilst venetoclax suppressed the interaction between Bcl-2 and Bak (Fig 1e and 1f). Flow cytometric analysis also verified that bort/venetoclax induced synergistic expression of cleaved caspase 3 (CC3) significantly in the sLCLs (Fig 1g and 1i). The cleavage of caspase 3 and PARP was inhibited upon co-treatment with caspase-9 inhibitor (Z-LEHD) and pan-caspase inhibitor (Z-VAD) but not caspase-8 inhibitor (Z-IETD). Meanwhile, caspase-9 was reduced or cleaved when the cells were co-treated with Z-IETD but remained intact when treated with Z-LEHD or Z-VAD. Together, the results suggested that the apoptosis induced by bort/venetoclax was likely to be activated through intrinsic pathway (Fig 1j).

## EBNA-3C, LMP-1 and bortezomib-induced Noxa in sLCLs were essential for the induction of cell death synergism by bortezomib/venetoclax

Lentivirus shRNA knockdown of protein expression of BHRF-1, EBNA-3C, LMP-1 or Noxa was performed to identify the role of the viral and host cell proteins for the cell death synergism induced by bort/venetoclax (S2 Fig). Three clones of lentiviruses (shBHRF1_1–3, shE3C_1–3 and shLMP1_1–3) for each knockdown of respective proteins were tested in sLCL 381 and sLCL 421. The knockdown efficiency was analyzed by western blotting (Fig 2a and

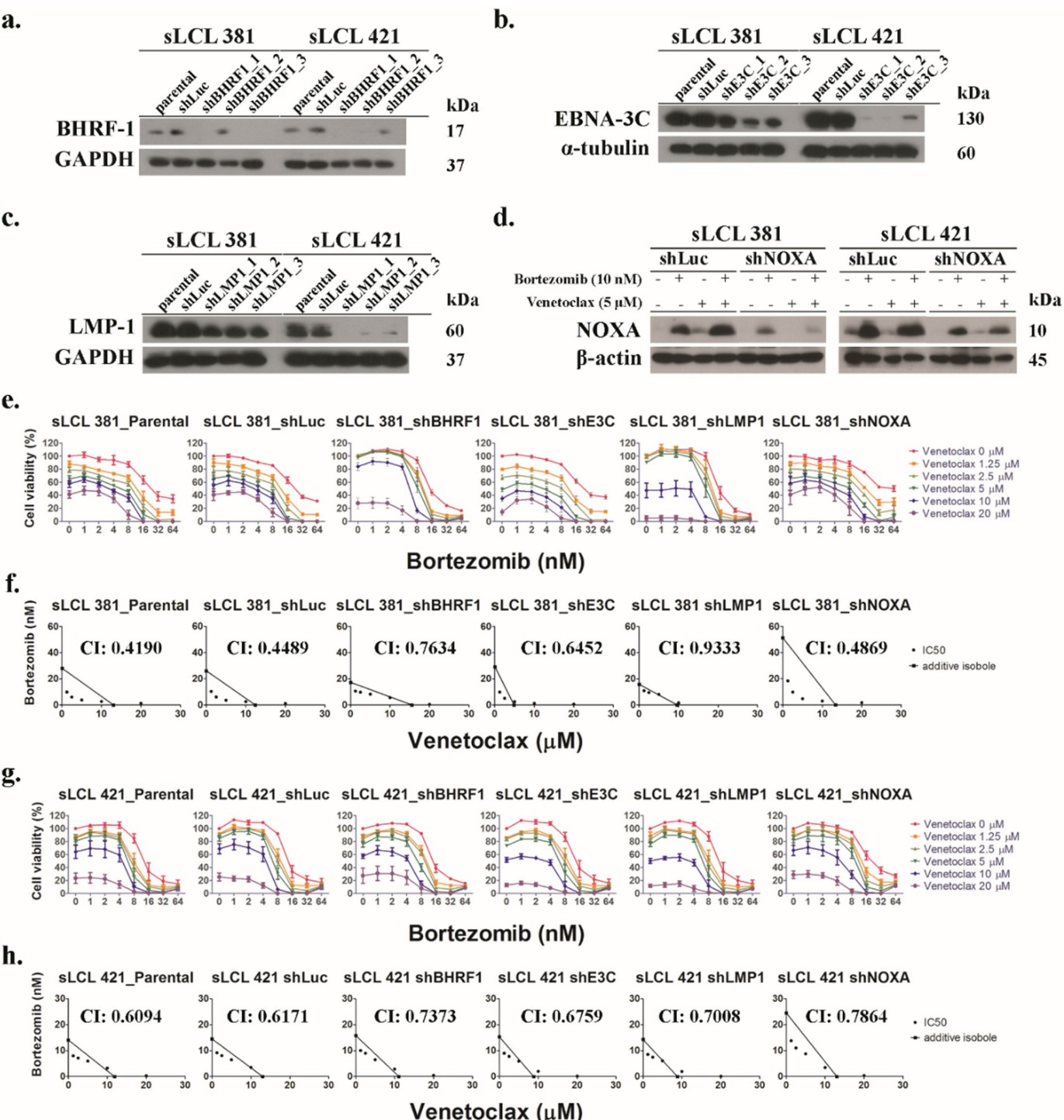

**Fig 2. Lentivirus shRNA knockdown experiment and MTT analysis showing the effect of bortezomib/venetoclax on the cell viability of sLCL 381 and 421 panels.** Lentivirus shRNA knockdown on (**a**) BHRF-1, (**b**) EBNA-3C, (**c**) LMP-1 and (**d**) Noxa were performed in sLCL 381 and 421 cells. Scramble shRNA on wild-type Firefly Luciferase (shLuc) was performed as a control. The expression of three different clones of BHRF-1, EBNA-3C and LMP-1were detected by western blot analysis. sLCL 381 and 421 shLuc and shNOXA were treated with conditions as described to test the expression of Noxa which was detected by western blot analysis. The cells were treated with combination of bortezomib (0, 1, 2, 4, 8, 16, 32, and 64 nM) and venetoclax (0, 1.25, 2.5, 5, 10, 20 μM) for 24 hours and stained by MTT solution. OD570 nm and OD630 nm were measured, and cell viability was plotted. (**e & g**) Percentages of cell viability of sLCL 381 and 421 panels in bort/venetoclax treatments were determined. (**f & h**) Isobolograms were applied for analysis of synergism of bort/venetoclax.

2c). The clones of shBHRF1_1, shE3C_2 and shLMP1_2 of sLCL 381 while those of shBHRF1_1, shE3C_1 and shLMP1_1 of sLCL 421 were used for this study, based on the knockdown efficiency and cell viability. sLCL 381 and sLCL 421 with shLuc and shNOXA were treated with either DMSO, 10 nM bortezomib, 5 μM venetoclax or bort/venetoclax for 24 hours to test the expression of Noxa which was reduced in shNOXA of both sLCLs (Fig 2d).

The relative cell proliferation and dose-response curve of either drug alone or combination were determined by MTT assays (Fig 2e and 2g). Knockdown of Noxa conferred the sLCLs modestly stronger resistance to bortezomib suggesting that induction of Noxa by bortezomib was indispensable to kill the sLCLs. The synergism of bort/venetoclax in these cell lines was further analyzed by isobologram analysis (Fig 2f and 2h). Combination indexes (CIs) of sLCL 381 parental, shLuc, shBHRF1, shE3C, shLMP1 and shNOXA were 0.4190, 0.4489, 0.7634, 0.6452, 0.9333 and 0.4869, respectively while those of sLCL 421 parental, shLuc, shBHRF1, shE3C, shLMP1 and shNOXA were 0.6094, 0.6171, 0.7373, 0.6759, 0.7008 and 0.7864, respectively. It was observed that CIs of shBHRF1, shE3C, shLMP1, and shNOXA of both sLCLs were increased when compared with those of their parental and scrambled control cell lines indicating that knockdown of BHRF-1, EBNA-3C, LMP-1 or Noxa led to the reduction in synergism of bort/venetoclax in the sLCLs.

Pairs of EBV-negative and EBV-positive BL2 and BL31 Burkitt's lymphoma cell lines were treated with bort/venetoclax to demonstrate the EBV specificity of the combination treatment. Bort/venetoclax elicited modestly stronger synergistic effect on EBV-positive than EBV-negative lines consistently (S7 Fig), which the CIs of EBV-negative BL2 and BL31 (BL2_EBV -ve:1.079, BL31_EBV -ve: 0.8668) were higher than the CIs of their EBV-positive counterparts (BL2_WT EBV: 0.9613, BL31_WT EBV: 0.7642). Moreover, the CIs of EBV positive BL2 and BL31 were higher than that of sLCL381 (0.4190) and 421 (0.6094), suggesting that bort/venetoclax had stronger synergism in sLCLs.

## Apoptosis induced by bortezomib/venetoclax was stronger in sLCLs expressing BHRF-1, EBNA-3C, LMP-1 and Noxa

TUNEL assay was performed to investigate the effects of bort/venetoclax on the induction of apoptosis in the above panels of cell lines of sLCL 381 and sLCL 421, (S3 Fig). Since the percentage of background apoptotic cells in each cell line was different as observed from the cells treated with DMSO only, the fold change of percentage of BrdU in each treatment of each cell line was normalized to DMSO control. The fold change of percentage of BrdU was significantly higher (***$p < 0.001$) in sLCLs, parental and shLuc than that in shBHRF1, shE3C, shLMP1 or shNOXA when the cells were treated with bort/venetoclax (Fig 3). The data confirmed that bort/venetoclax could induce stronger apoptosis in sLCLs expressing BHRF-1, EBNA-3C, LMP-1 and Noxa.

Western blotting showed that Bcl-6 and IκBα were increased in the sLCLs upon treatment with bortezomib. Moreover, the phosphorylation of NF-κB p65 were synergistically reduced by bort/venetoclax in shLuc and shE3C cells (Fig 4a). In addition, a marked decrease of Bcl2 and p-Bcl2 in shE3C and shLMP1 cells when compared to the level of Bcl2 in parental, shLuc or other knockdown cells (Fig 4b and 4c) indicated that EBNA3C or LMP1 drove Bcl2 expression in sLCLs. This finding was in line with that of previous studies [22–24,31]. The decrease in Bcl-2 expression might give rise to the decrease in synergism of bort/venetoclax in the sLCLs with shE3C and shLMP1 (Fig 2e–2h).

Furthermore, weaker p-Bcl-2 (Ser70) accompanied with stronger CC3 and higher c-PARP/total-PARP ratio in the sLCLs parental and shLuc than in shBHRF1, shNOXA, shLMP1 and shE3C treated with bort/venetoclax was demonstrated. Meanwhile, p-H2AX (Ser139) and p-

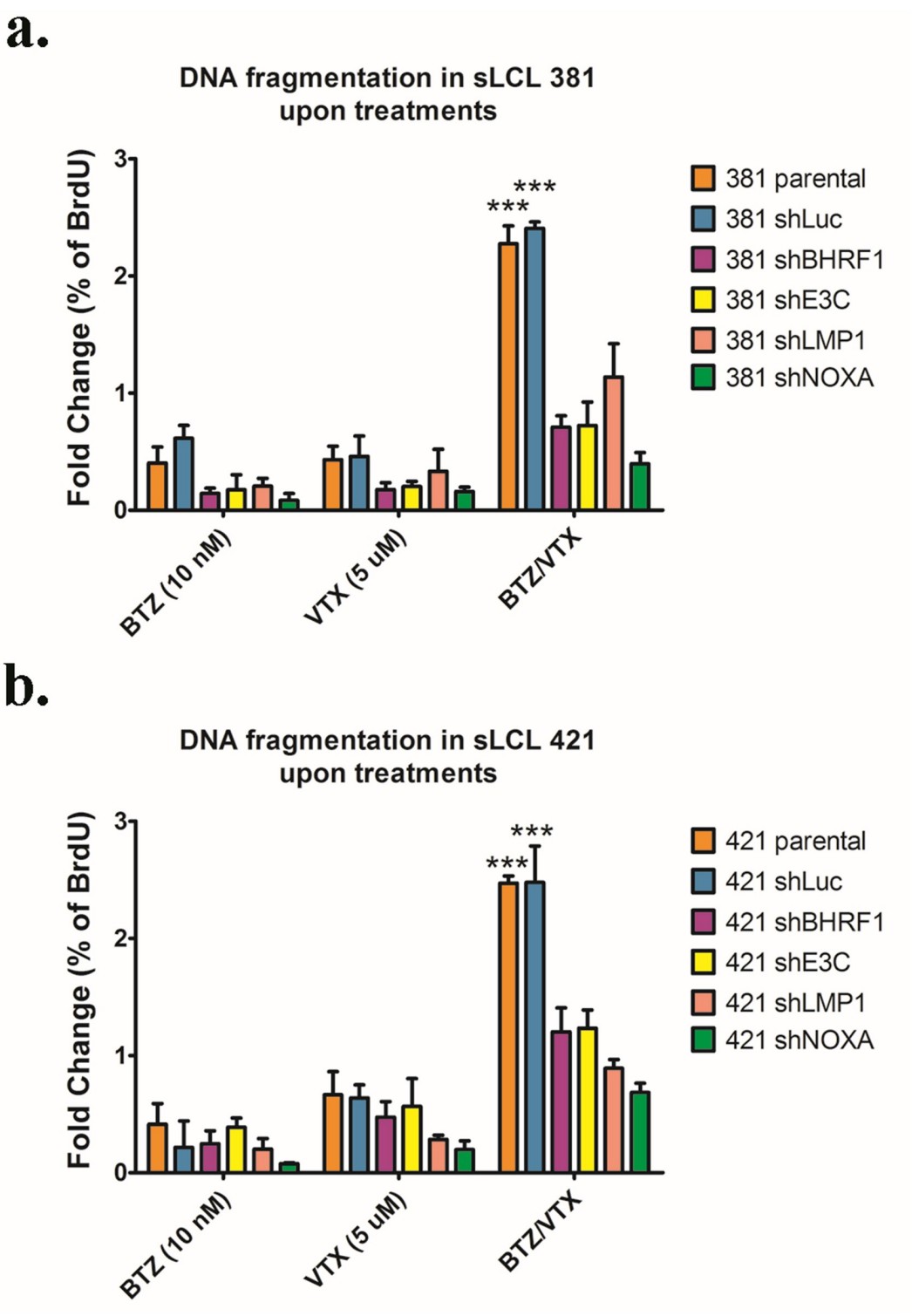

**Fig 3. Apoptosis induced by bortezomib/venetoclax was stronger in sLCL expressing BHRF-1, EBNA-3C, LMP-1 and Noxa.** **(a)** sLCL 381 and **(b)** sLCL 421 panels were were either treated with DMSO, 10 nM bortezomib (BTZ), 5 μM venetoclax (VTX) or bortezomib/venetoclax (BTZ/VTX) for 24 hours. The cells were stained by TUNEL as described and analyzed by flow cytometry. Fold change of percentage of BrdU in the cells upon treatments was calculated. The fold change was analyzed for statistical significance by using One-way ANOVA Bonferroni post-tests. *p < 0.05 was considered as statistically significant (***p < 0.001).

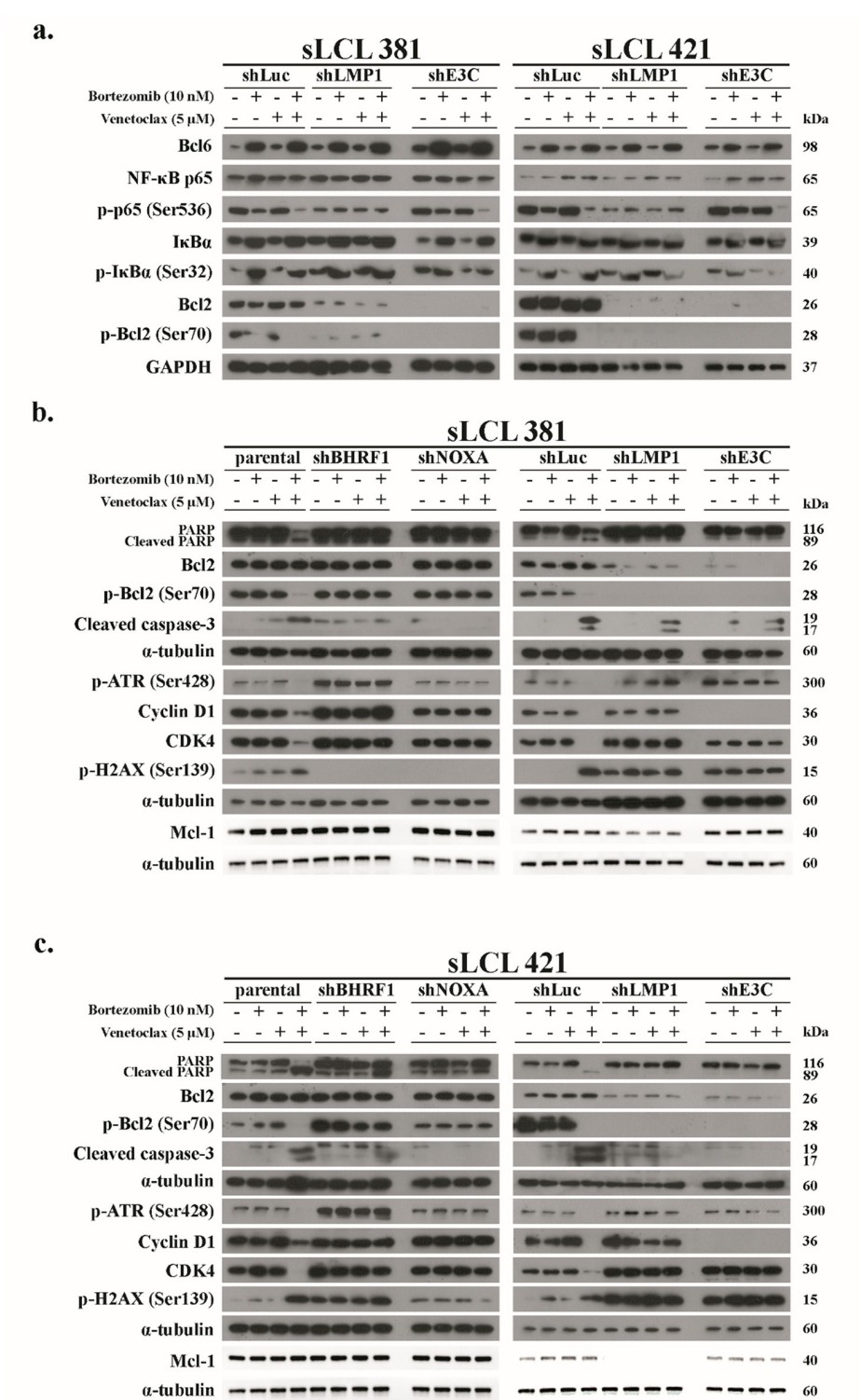

**Fig 4. Synergistic apoptosis, DNA damage response and inhibition of NF-κB induced by bortezomib/venetoclax was related to the expression of BHRF-1, LMP-1, EBNA-3C and botezomib-induced Noxa in sLCL.** (**a**) shLuc, shLMP-1 and shE3C of sLCL 381and sLCL 421 were treated with conditions as described. Proteins were extracted and subjected to western blot analysis with indicated antibodies. (**b**) sLCL 381 and (**c**) sLCL 421 panels were treated with conditions as described. Proteins were extracted and subjected to western blot analysis with indicated antibodies.

ATR (Ser428) were relatively stable in shBHRF1, shNOXA, shLMP1 and shE3C sLCLs suggesting that potent DDR and apoptosis induced by bort/venetoclax were caused by the expression of BHRF-1, LMP-1, EBNA-3C and bortezomib-induced Noxa. The expression of Mcl-1 was decreased only in shLMP1 sLCLs verifying that LMP-1 promoted the expression of Mcl-1 through NF-κB signaling pathway [20] (Fig 4b and 4c).

## Bortezomib/venetoclax suppressed G1-S transition in sLCLs expressing EBNA-3C, LMP-1 and bortezomib-induced Noxa

The expression of cyclin D1 and CDK4 was reduced in parental and shLuc sLCLs whereas no change in the level of these proteins in shBHRF1, shNOXA, shLMP1 or shE3C sLCLs upon treatment of bort/venetoclax was observed (Fig 4b and 4c). Cell cycle analysis verified that either bortezomib or venetoclax induced slight G1 arrest whereas bort/venetoclax induced much stronger suppression of G1-S transition in the parental and shLuc sLCLs. Nevertheless, such G1 arrest was weaker or not observed in either shBHRF1, shE3C, shLMP1 or shNOXA suggesting that the G1-S transition was facilitated by EBNA-3C and LMP-1. Moreover, bortezomib-induced Noxa was also vitally important to achieve the G1 arrest (Figs 5, S4a and S4b). Bort/venetoclax yielded greater percentage of sub-G1 populations in parental and shLuc sLCLs denoting stronger apoptosis in these cells (S4c and S4d Fig). Taken together, these data demonstrated that the expression of EBNA-3C and LMP-1 could promote G1-S transition in the sLCLs whilst bort/venetoclax and bortezomib-induced Noxa promoted G1 arrest and apoptosis.

## Bortezomib/venetoclax synergistically inhibited the growth sLCL xenografts in vivo

The effect of bort/venetoclax was further evaluated *in vivo* on the growth of sLCL xenografts in SCID mice. On day 30, the average tumor volume in the groups of mice treated with vehicle control, bortezomib, venetoclax or bort/venetoclax was 263.2 mm$^3$, 132.8 mm$^3$, 129.9 mm$^3$ and 19.1 mm$^3$, respectively, whilst the average tumor mass in the corresponding groups was 95.7 g, 71.03 g, 71.43 g and 8.017 g, respectively (Fig 6a and 6b). When compared with bortezomib or venetoclax alone, bort/venetoclax resulted in much stronger inhibition of the tumor growth (Fig 6c). One tumor was completely regressed in a mouse treated with bort/venetoclax (S5a and S5b Fig). EBER in-situ hybridization and CD20 immunohistochemical (IHC) staining of the sections of the xenografts validated that the tumor cells expressed EBER and CD20. (S5c Fig). The result of phospho-H2AX (pH2AX) IHC staining showed that the number of positively stained cells per high power field of combination treatment was significantly higher than vehicle and single treatment. Since significant difference in tumor regression was also observed in combination than single treatment (Fig 6a and 6b), the quantitation of pH2AX positive cells seems to correlate (Fig 6e and 6f). Toxicity of the treatment in the mice was assessed by monitoring their weight (Fig 6d). The data suggested that the synergistic killing of sLCL by bort/venetoclax could also be observed *in vivo*.

## Discussion

Current treatment of post-transplant lymphoproliferative disorder (PTLD) includes multi-agent chemotherapy, rituximab and adoptive immunotherapy. Chemotherapy has relatively high toxicity and is poorly tolerated in immunocompromised patients. Rituximab, a chimeric anti-CD20 monoclonal antibody, is better tolerated. However, it is only effective in the benign spectrum of PTLD such as polymorphic PTLD but not in monomorphic PTLD. Though

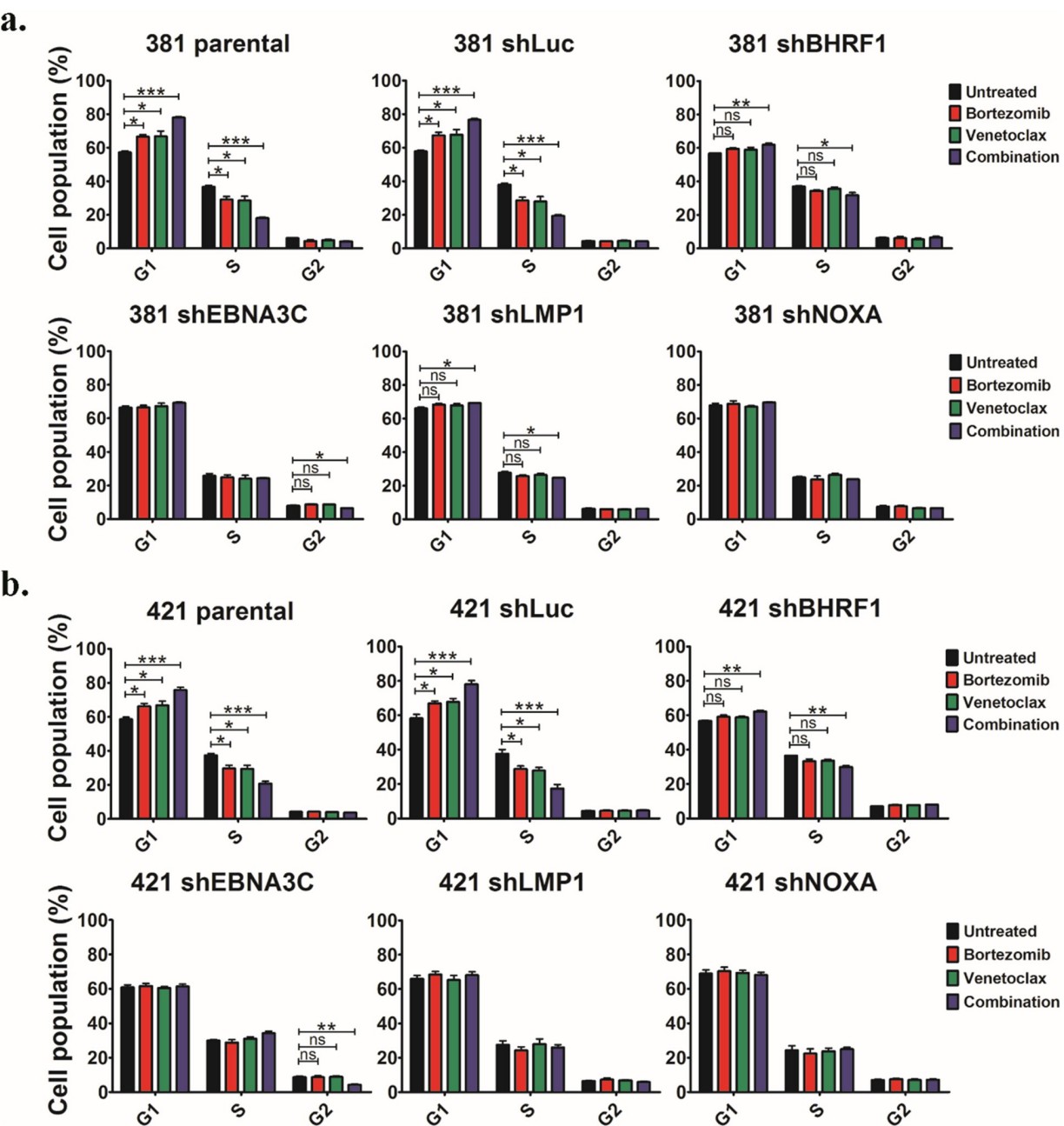

**Fig 5. Bortezomib/venetoclax suppressed G1-S transition in sLCLs expressing EBNA-3C, LMP-1 and bortezomib-induced Noxa.** sLCL 381 and sLCL 421 panels were treated with conditions as described. The treated cells were subjected to cell cycle analysis as described and analyzed by ModFit LT 3.0. **(a-b)** Cell cycle analysis showing the percentages of sLCL 381 and 421 cells in G1, S, G2/M phases. The results were analyzed for statistical significance using One-way ANOVA Dunnett's Multiple Comparison Test. p-value less than 0.05 was considered statistically significant; *p < 0.05, **p < 0.01, ***p < 0.001, ns = not significant. Error bars represent the standard error of mean (SEM) of data obtained from three independent experiments.

adoptive immunotherapy is an effective treatment method, the cost, time of production and need for specialized production facility are limiting its wider clinical application. Here, we reported that the combination of bortezomib and venetoclax (bort/venetoclax) synergistically killed lymphoblastoid cell lines (LCLs) by targeting the pro-survival function of EBV latent

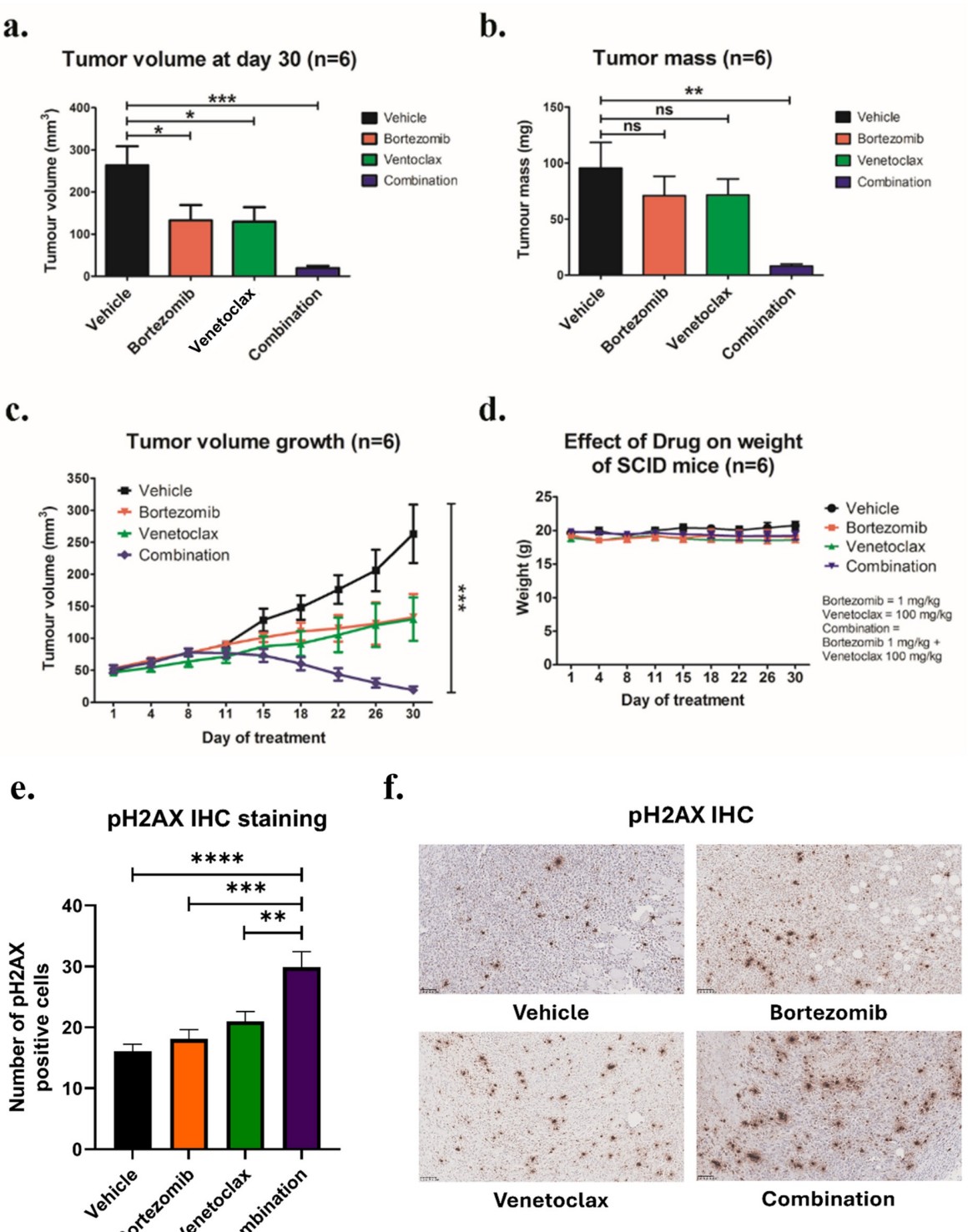

**Fig 6. Effects of bortezomib/venetoclax on the growth suppression of sLCL 381 xenografts in SCID mice.** SCID mice were engrafted with sLCL 381 xenograft and treated with conditions as described (n = 6 for each treatment group). **(a)** The average of tumor volumes on day 30 and **(b)** the average of tumor masses of mice of control and treated groups were shown. **(c)** The size of tumors during the period of experiment was measured twice weekly using a caliper. Data are presented as the mean tumor volumes of mice in both treatment and control groups on the day of treatment (day 1, 4, 8, 11, 15, 18, 22, 26 and 30). **(d)** The mice were weighed on day 1, 4, 8, 11, 15, 18, 22, 26 and 30 to assess the toxicity of the treatments. **(e)** pH2AX IHC staining of the xenografts were performed as described. The number of

pH2AX positive cells were summarized in histograms and (**f**) a representative IHC image of each treatment group was shown. The results were analyzed for statistical significance using One-way ANOVA Dunnett's Multiple Comparison Test. p-value less than 0.05 was considered statistically significant; *p < 0.05, **p < 0.01, ***p < 0.001, ****p < 0.0001 and ns = not significant. Error bars represent the standard error of mean (SEM) of data obtained from the SCID mice.

proteins including latent membrane protein 1 (LMP-1) and EBV nuclear antigen 3C (EBNA-3C) and resulted in synergistic induction of DNA damage followed by G1-S cell cycle arrest and apoptosis. Detailed mechanisms of action of the drug combination were delineated. Bortezomib potently induced the expression of Noxa, a pro-apoptotic initiator and when combined with venetoclax, inhibited Mcl-1 and Bcl-2 simultaneously as well as prevented LMP-1-induced proteasomal degradation of IκBα leading to the suppression of the NF-κB signaling pathway. Bortezomib also rescued Bcl-6 from EBNA-3C-mediated proteasomal degradation thus maintaining the repression of cyclin D1 and Bcl-2 causing G1-S arrest and apoptosis. Concurrently, venetoclax inhibited Bcl-2 upregulated by either LMP-1 or EBNA-3C (refer to Fig 7).

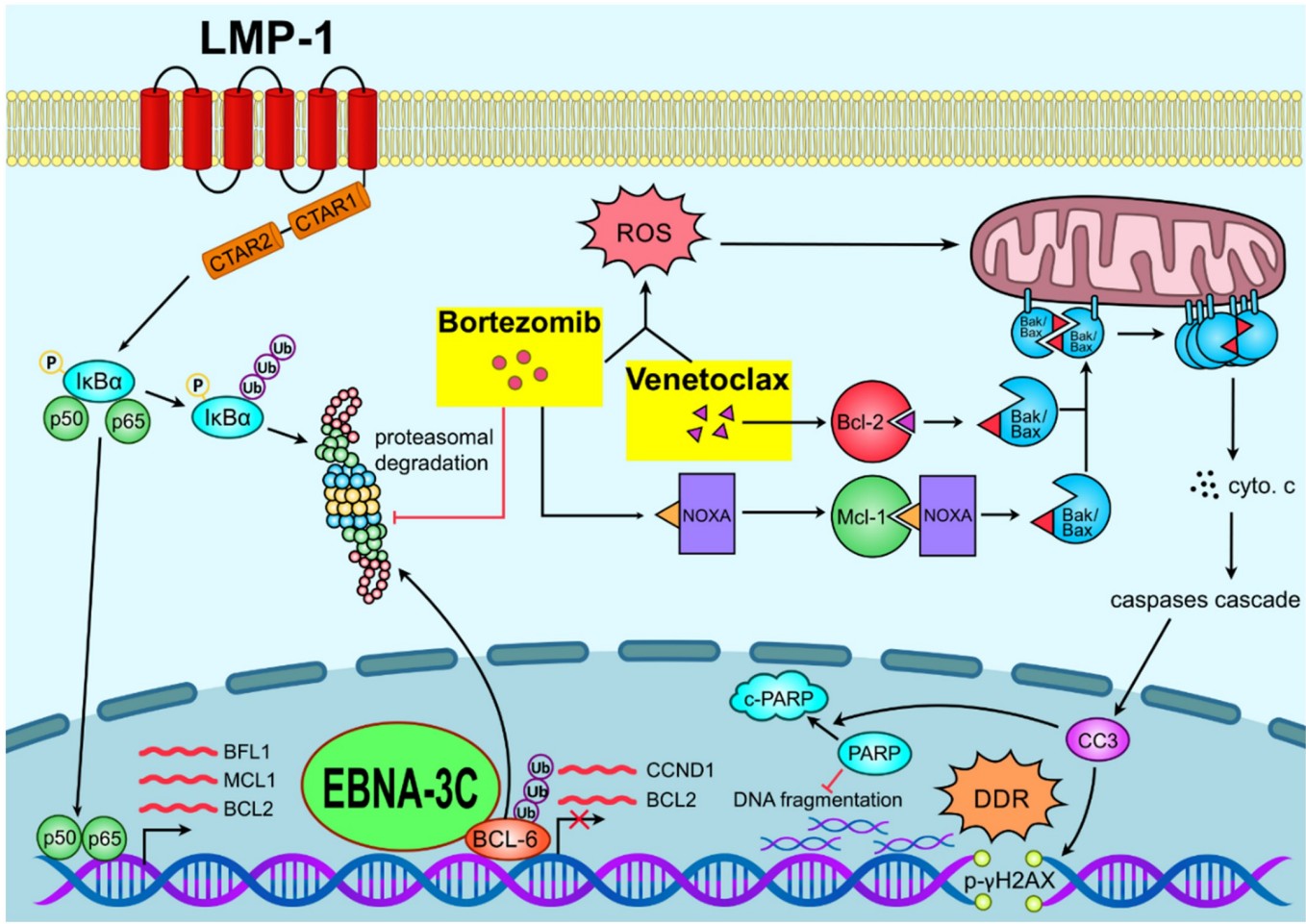

**Fig 7. Schematic diagram illustrating the mechanism of action of bortezomib/venetoclax in sLCLs.**

We observed that the protein level of Bcl-2 was not reduced. This implied that the expression of Bcl-2 might also be regulated by other transactivators such as cyclic adenosine monophosphate-responsive element binding proteins [32] and B-Myb [33]. Despite the unchanged level of Bcl-2 protein upon treatment with bort/venetoclax, phosphorylated-Bcl-2 (p-Bcl-2) at Serine 70 was significantly decreased in parental and shLuc-transduced sLCL 381 and 421. p-Bcl-2 (Ser70) enhanced the anti-apoptotic function of Bcl-2 through stronger interaction with Bak and Bax [34,35] whilst phosphomimetic of Bcl-2 (Ser70) rendered 100 to 300-fold less binding affinity to BH3-mimetics including venetoclax and prevented cytochrome c release [36]. As a result, the reduced p-Bcl-2 (Ser70) induced by bort/venetoclax might be paramount to inducing the potent apoptosis.

The development of EBV-associated PTLD is the consequence of reduced immunosurveillance on EBV antigens due to immunosuppression. Hence, immune control against EBV is critical and whether bort/venetoclax has immunosuppressive effect is a relevant question. Preclinical studies on bortezomib and venetoclax and T cell function demonstrated that both drugs could sustain T cell function. Renrick et al. [37] treated BALB/c WT mice with bortezomib at 1 mg/kg and purified CD8+ T-cells from a pool of spleen and lymph nodes and reported that the expression of miR-155 was induced. The increased miR-155 led to the down-regulation of its target protein SOCS1 and SHIP1 which are T-cell negative regulatory proteins. Moreover, a decrease in programmed cell death-1 (PD-1) expressing SHIP1+ phenotype was observed in activated CD8+ T cells. Collectively, bortezomib has stimulatory effects on CD8+ T cells. Kohlhapp et al. [38] reported that venetoclax did not affect antigen-specific functional human T-cell responses. When cytomegalovirus (CMV)-specific CD8+ T cells from human peripheral blood mononuclear cells of CMV-positive carriers were stimulated by CMV peptides together with increasing concentration of venetoclax, a decrease in cell number but similar amount of IFNγ production were observed. Furthermore, healthy human subjects treated with 100mg of venetoclax had increase in the fraction of CD4+ and CD8+ effector memory cells [38].

Bort/venetoclax could also potentially mitigate the side effects associated with either drug alone. Bortezomib might reduce venetoclax-induced tumor lysis syndrome, mild gastrointestinal side effects and neutropenia [39] whilst venetoclax could minimize the adverse effects of bortezomib including diarrhea, neutropenia, thrombocytopenia and peripheral neuropathy [40]. A recent phase I clinical study assessed the safety of venetoclax combined with daratumumab and dexamethasone with (VenDdB) or without bortezomib (VenDd) in myeloma patients and demonstrated that fewer patients treated with VenDdB experienced ≥ grade 3 AEs than those treated with VenDd [41].

We employed lentivirus shRNA knockdown experiments to suppress the expression of either BHRF-1, LMP-1 or EBNA-3C in the two sLCLs. Of note, the cell death synergism induced by bort/venetoclax in the sLCLs was decreased when the expression of these proteins was individually suppressed suggesting that bort/venetoclax targeted the pro-survival function of BHRF-1, EBNA-3C or LMP-1 resulting in synergistic killing of the sLCLs. We postulated that bort/venetoclax might have a wider clinical application for other EBV-associated lymphoma dependent on LMP-1, EBNA-3C or BHRF-1 for survival function and be particularly beneficial to patients with immunodeficiency-associated lymphoma or lymphoma associated with aging such as EBV-positive diffuse large B cell lymphoma. Approximately 1 in 10 cases of diffuse large B cell lymphoma (DLBCL) is EBV-positive in Asian countries [42,43] and EBV positivity contributed to poorer prognostic impact than EBV-negative DLBCL with worse overall survival and progression-free survival [44]. EBV-positive DLBCL has latency II/ III pattern amongst which LMP-1 is expressed [45]. Huang et al. [46] showed that EBV infection was associated with reduced EphA4 expression which could be a biomarker that correlated with

poor survival outcome in DLBCL. LMP-1 could trigger Sp1 level via ERK pathway to suppress EphA4 promoter activity lending evidence of pro-survival function of LMP-1 in EBV-positive DLBCL. Rituximab plus chemotherapy (R-CHOP) has been the standard of care for DLBCL. However, dose reduction or premature treatment discontinuation was often required due to toxicity in the immunocompromised or aged patients [47]. We predict that bort/venetoclax would be able to target the pro-survival function of LMP-1 in DLBCL and better tolerated by the aged patients.

In conclusion, our results revealed a novel synergistic action of bortezomib combined with venetoclax in targeting lymphoblastoid cells through abrogating the pro-survival function of LMP-1 and EBNA-3C. The induction of DNA damage response and inhibition of DNA repair was followed by G1-S cell cycle arrest and apoptosis. Combined regimen of bortezomib and venetoclax specifically targets the cell survival vulnerabilities of EBV-driven lymphoproliferative diseases such as PTLD and has potential clinical application for a wider spectrum of EBV-driven lymphomas.

## Materials and methods

### Ethics statement

The collection of the new samples received approval from the Institutional Review Board of The University of Hong Kong/Hospital Authority Hong Kong West Cluster (UW 21–351). Written informed consents were obtained from the participants and/or their guardian.

### Cells and culturing conditions

sLCL 381, sLCL 366, sLCL 401 and sLCL 478 were derived from clinically diagnosed post-transplant lymphoproliferative disorder (PTLD) patients, which sLCL 381 was derived from the patient having T cell PTLD. sLCL 397 and sLCL 421 were derived from clinically diagnosed hemophagocytic lymphohistiocytosis (HLH) patients. The sLCLs were established from spontaneous outgrowth of blood mononuclear cells of bone marrow of the paediatric patients. PrimeFlow RNA assay (Thermo Fisher Scientific, USA) confirmed that these two sLCLs are B cells infected with EBV (S1 Fig). sLCL 381 (47, XY, +12) and sLCL 421 (47, XY, +12) were karyotyped by Laboratory of Haematology Cytogenetics (Hong Kong Children's Hospital). All the sLCLs were maintained in RPMI 1640 culture medium supplemented with 15% FBS and 1% penicillin-streptomycin (P/S) and grown in a humidified incubator at 37°C with 5% $CO_2$. Parental BL2 (BL31 EBV -ve) and BL31 (BL31 EBV -ve) are EBV-negative Burkitt's lymphoma (BL) cell lines. BL2 EBV-WT and BL31 EBV-WT are EBV-positive BL cell lines (obtained from Prof. M. Allday, Imperial College, London, UK). All BL cell lines were maintained in RPMI 1640 culture medium supplemented with 10% fetal bovine serum (FBS), 1 mM sodium pyruvate, 50 mM a-thioglycerol and 1% penicillin and streptomycin. 100 mg/ml hygromycin B was used in culture of EBV positive BL cell lines.

### Drug treatment

Unless otherwise specified, cells were seeded at a density of 1 x $10^6$/ ml 24 hours prior treatment and were treated with DMSO, 10 nM bortezomib, 5 μM venetoclax or combination for 24 hours in a 5% $CO_2$, 37°C incubator. Chemical compounds used: bortezomib (Selleck Chemicals, Houston, TX, USA), SAHA (Cayman Chemicals, Ann Arbor, MI, USA) and venetoclax (AdooQ BioScience, California, USA).

## Lentivirus short-hairpin RNA (shRNA) knockdown

shRNA targeting shLuc (5'-GTGCGTTGTTAGTACTAATCCTATTT-3'), shBHRF1 (5'-GTGTTGCTTGAGGAGATAATT-3'), shLMP1 (5'-GACCTCATCCTGCTCATTATT-3'), shE3C (5'-CCATATACCGCAAGGAATA-3') and shNOXA (5'- GGTGCACGTTTCAT-CAATTTG-3') were cloned into pGreenPuro plasmid (System Biosciences) with BamHI and EcoRI restriction sites and the loop GTGAAGCAGATG was in between the sense and anti-sense strand. Plasmids were verified with Sanger sequencing service (Centre for PanorOmic Sciences (CPOS), Genomics Core, LKS Faculty of Medicine, The University of Hong Kong). These plasmids and lentivector pPACKH1 packaging plasmids (System Biosciences) were then transfected into 293T cells at 70–80% confluence by GeneJuice (Sigma-Aldrich) with an incubation time of 72 hours. The medium was collected and centrifuged at 3000 x g at room temperature for 15 minutes to pellet cell debris. The lentiviral supernatant was filtered through 0.45-μm-pore-size cellulose acetate filters, mixed with cold PEG-it Virus Precipitation Solution (System Biosciences) at 4:1 ratio, and refrigerated overnight. The next day, the supernatant/PEG-it mixture was centrifuged at 1500 x g for 30 minutes at 4˚C. After centrifugation, the lentivector particles appear as a beige or white pellet at the bottom of the vessel. The supernatant was removed, and the pellet was resuspended in cold Opti-MEM. The titre of the lentivirus was determined by following the protocol of Dharmacon GIPZ Lentiviral shRNA. The lentivirus was transduced to sLCL 381 and 421 at multiplicity of infection (MOI) of 2. The cells were maintained in RPMI 1640 culture medium supplemented with 15% FBS and 1% P/S and selected by 2 μg/mL puromycin for two weeks.

## Western blot analysis

Protein from the treated cell cultures was extracted and western blot analysis was performed as described previously [48]. Expression of target proteins were detected with antibodies listed in (S11 Fig).

## Co-immunoprecipitation

Co-immunoprecipitation was performed by using Dynabeads Co-Immunoprecipitation kit (Invitrogen, USA) and following manufacturer's instructions. Briefly, anti-Bak antibody Y164 (Abcam, UK) was coupled to Dynabeads, followed by incubation with 90% of total protein lysate extracted from cells and then eluted as IP:Bak portion. 10% of the total protein lysate was used as input. Western blot analysis was performed as described previously [48].

## MTT assay

sLCL 381 and 421 panels (parental, shLuc, shBHRF1, shLMP1, shE3C and shNOXA) cells were treated with combination of bortezomib (0, 1, 2, 4, 8, 16, 32, and 64 nM) and venetoclax (0, 1.25, 2.5, 5, 10, 20 μM) for 24 hours. 10 μL MTT solution was added into each well and wells were incubated in a 37˚C, 5% $CO_2$ incubator for 5 hours. OD630 nm and OD570 nm were measured and cell viability was plotted. SynergyFinder, isobolograms analysis and combination index (CI) were used to evaluate the synergistic action of drug combination. The data of cell viability was loaded to SynergyFinder [49], a web-app (https://synergyfinder.fimm.fi), for computing synergy score. Synergy score less than -10, between -10 and 10, and greater than 10 indicated antagonism, additivity, and synergism respectively. Isobolograms analysis and combination index (CI) were calculated using the Chou and Talalay method [50]. Isobologram were generated from the different combinations of concentrations of each drug which inhibit 50% of the cells' proliferation ($IC_{50}$). Isoboles (black dots) for $IC_{50}$ that were located to

the left or right of the additive isoboles (oblique lines) indicated synergistic or antagonistic action respectively. CI < 1, = 1, and >1 indicated synergism, additivity, and antagonism respectively.

The following equation was used to calculate CI:

$$CI = \frac{[\text{Drug A}] \text{ of IC50 in combination}}{[\text{Drug A}] \text{ of IC50 alone}} + \frac{[\text{Drug B}] \text{ of IC50 in combination}}{[\text{Drug B}] \text{ of IC50 alone}}$$

## Cell cycle analysis

sLCL 381 and 421 panels (parental, shLuc, shBHRF1, shLMP1, shE3C and shNOXA) cells were either treated with DMSO, 10 nM bortezomib, 5 μM venetoclax, or combination of 10 nM bortezomib and 5 μM venetoclax. After 24 hours incubation, cells were first washed with PBS once and were subsequently fixed with 70% ethanol overnight at −20˚C. The next day, the cells were washed with PBS once and were resuspended and incubated in PBS supplemented with 500 μg/ml RNase for 10 minutes at room temperature. The cells were then stained with 50 μg/ml of propidium iodide (Invitrogen, USA) in dark at room temperature for 15 minutes. After staining, the cells were subjected to cellular DNA content analysis by a flow cytometer (LSRII, BD Biosciences). Data were analysed by ModFit LT 3.0 software.

## Terminal deoxynucleotidyl transferase–mediated dUTP nick end labeling (TUNEL) assay

sLCL 381 and 421 panels (parental, shLuc, shBHRF1, shLMP1, shEC and shNOXA) cells were either treated with DMSO, 10 nM bortezomib, 5 μM venetoclax, or combination. Cells were collected at 24 hours post-treatment and washed twice with PBS. The cells were subsequently fixed with 10% neutral buffered formalin (NBF) for 15 minutes and washed twice with PBS. The cells were then fixed with 70% ethanol overnight at −20˚C. The next day, the cells were washed with PBS twice. TUNEL staining was then conducted with APO-BrdU TUNEL Assay Kit (Invitrogen, USA) following manufacturer's instructions. The deoxythymidine analog 5-bromo-2′-deoxyuridine 5′-triphosphate (BrdU) was stained by APC-conjugated anti-BrdU antibody (eBioscience). The stained cells were measured by flow cytometry (LSRII; BD Biosciences) and data were analyzed by FlowJo software (Tree Star).

## RNA and RNA-seq library preparation

sLCL 381 and sLCL 421 cells were treated with either DMSO, 10 nM bortezomib, 5 μM venetoclax or combination for 24 hours. Total RNA was extracted by using TRIzol (Invitrogen, USA) according to the manufacturer's instructions. RNA quality was evaluated using a bioanalyzer. RNA integrity numbers (RINs) were > 9, except samples of sLCL 421 treated with combination, which ranges from 7.7 to 8.3. Library preparation and Illumina sequencing (Pair-End sequencing of 151bp) were done at Centre for PanorOmic Sciences (CPOS), Genomics Core, LKS Faculty of Medicine, The University of Hong Kong. 100 ng of total RNA was used as starting material and cDNA libraries were prepared by KAPA Stranded mRNA-Seq Kit according to manufacturer's protocol. Illumina NovaSeq 6000 was used for Pair-End 151bp sequencing. Using software from Illumina (bcl2fastq), sequencing reads of the samples ranges from 61 to 72 million. An average of 94% of the bases achieved a quality score of Q30 where Q30 denotes the accuracy of a base call to be 99.9%.

### RNA-seq data analysis

Raw data analysis was performed by Centre for PanorOmic Sciences (CPOS), Genomics Core, LKS Faculty of Medicine, The University of Hong Kong. Sequencing reads were first filtered for adapter sequence and low quality sequence followed by retaining only reads with read length $\geq$ 40bp. Sequencing reads were then filtered for rRNA sequence and remaining reads were mapped to the reference genome (Human Genome GRCh38 (downloaded from GEN-CODE)) using STAR (default parameters). Differentially expression analysis was done using EBSeq and Differentially expressed genes were defined as false discovery rate (FDR) < 0.05.

### Pathway enrichment analysis

Differentially expressed genes with FDR < 0.05 were adjusted for multiple testing comparisons using Bonferroni's correction. Pathway enrichment was analyzed by Fisher's Exact test using Partek Genomics Suite 6.6.

### In vivo experiment

$1 \times 10^7$ of sLCL 381 cells were resuspended in 100 µl of PBS and subcutaneously injected at the right flanks of Female C.B-17/Icr-scid (SCID) mice, at age of 6–7 weeks. When the tumors reached to a mean volume of approximately 50 mm$^3$, the mice were treated with either DMSO (vehicle control, n = 6), 1 mg/kg bortezomib (n = 6), 100 mg/kg venetoclax (n = 6), or bortezomib/venetoclax (n = 6). Three doses of bortezomib (1 mg/kg) or an equal volume (100 ul) of vehicle was injected intraperitoneally (IP) at day 1, 5 and 9. Venetoclax or an equal volume (100 ul) was administered via oral gavage (PO) for 5 days per week until day 22. The mice were observed for an additional week after the end of treatment and then euthanized by 150 mg/kg 0.6% pentobarbital via IP injection. The size of the tumors and weight of the mice were measured as described previously [12]. All experimental procedures were reported and approved by Committee on the Use of Live Animals in Teaching and Research (CULATR) of the University of Hong Kong (CULATR 4135–16).

### Histological analysis

After euthanasia, the mouse tumors were dissected and fixed immediately in 10% neutral buffered formalin at room temperature for at least 24 hours. The tumors were then embedded in paraffin, cut into sections of 4 µm in thickness and mounted on slides which were stained with hematoxylin and eosin for microscopic examination. Immunohistochemistry (IHC) of CD20 was performed by using anti-CD20 antibody (clone L26, Dako, dilution 1:2400) and automated IHC stainer (Ventana BenchMark ULTRA, Roche Diagnostics, Switzerland) according to the manufacturer's instructions. IHC of phospho-H2AX (Ser139) was performed by using anti-pH2AX antibody (clone 20E3, Cell Signaling Technology, dilution 1:50) and automated IHC stainer (Leica BOND III, Leica Biosystems, Germany) according to the manufacturer's instructions. In each treatment group, at least 3 tumor sections were counted for pH2AX positive cells (Vehicle n = 5, Bortezomib n = 5, Venetoclax n = 5, Combination n = 3). Within each tumor section, three high power fields (circle of 0.55 mm diameter) in hotspot areas of pH2AX expression were selected and the number of pH2AX strongly positive cells in each circular field were counted. EBV status of the tumor was confirmed by in-situ hybridization (ISH) using Ventana INFORM EBV-encoded small RNAs (EBER) probe (Roche Diagnostics, Switzerland) and automated ISH stainer (Ventana BenchMark ULTRA). The histology slides were scanned using Hamamatsu Nanozoomer S210 (Hamamatsu Photonics, Japan), and photomicrographs were captured using software provided by the manufacturer.

## Statistical analysis

All *in vitro* experiments were performed in at least triplicate independently. Data here are represented as mean values with standard errors of means (SEM). Data were analyzed for statistical significance using One-way ANOVA Dunnett's Multiple Comparison Test. $^*$p-value $< 0.05$, $^{**}$p-value $< 0.01$ and $^{***}$p $< 0.001$ were considered as statistically significant, strongly significant, extremely significant, respectively. All statistical analyses were performed with GraphPad Prism Version 6.0 software.

## Supporting information

**S1 Fig. PrimeFlow RNA assay verified that the sLCL 381 cells are EBV-infected B cells.** 1.5 x $10^6$ **(a)** sLCL 381 and **(b)** sLCL 421 cells were stained by Aqua Blue dye (Invitrogen, USA) and the following fluorochrome-conjugated antibodies: Pacific Blue anti-human CD3, PE-Cy7 anti-human CD8, PE anti-human CD19, FITC anti-human CD16/56 (BioLegend, USA) and PE-Texas Red anti-CD4 (eBioscience, USA) on ice for 30 minutes for indication of cell viability and immunophenotype. The cells were fixed by fixation buffer and subsequently treated with PrimeFlow RNA permeabilization buffer with RNase inhibitors. The EBER in the cells were stained by fluorochrome-conjugated probe EBER-AF647 (Thermo Fisher Scientific, USA) and the cells were incubated in the dark at 4˚C overnight. On the next day, the cells were incubated with PrimeFlow RNA PreAmp Mix, PrimeFlow RNA Amp Mix, and diluted Label Probes for signal amplification. The samples were measured by flow cytometry and analyzed by FlowJo software (Tree Star).
(TIF)

**S2 Fig. sLCL 381 and 421 panels were observed by fluorescence microscopy (10X). (a)** sLCL 381 panel and **(b)** sLCL 421 panel transduced with lentivirus of either shLuc (scrambled control), shBHRF1, shE3C, shLMP1 or shNOXA and under puromycin selection (2 µg/ml) for at least 8 weeks were illustrated. Parental cells mean the sLCLs without transduction of lentivirus.
(TIF)

**S3 Fig. Apoptosis induced by bortezomib/venetoclax was stronger in sLCL expressing BHRF-1, EBNA-3C, LMP-1 and NOXA. (a)** sLCL 381 and **(b)** sLCL 421 panels were either treated with DMSO, 10 nM bortezomib, 5 µM venetoclax or bortezomib/venetoclax for 24 hours. The cells were harvested and fixed by 10% neutral buffered formalin (NBF) and 70% ethanol at -20˚C overnight. The DNA break was labelled by TdT and BrdU, which was detected by APC-conjugated anti-BrdU antibody and measured by flow cytometry. One representative set of TUNEL assays of the cell lines upon treatments was presented.
(TIF)

**S4 Fig. G1 cell cycle arrest and sub-G1 population induced by bortezomib/venetoclax were stronger in sLCL expressing BHRF-1, EBNA-3C, LMP-1 and NOXA. (a)** sLCL 381 and **(b)** sLCL 421 panels were either treated with DMSO, 10 nM bortezomib, 5 µM venetoclax or bortezomib/venetoclax for 24 hours. The next day, the cells were incubated with 500 µg/ml RNase for 10 minutes, followed by propidium iodide (PI) staining. The cellular DNA content was measured by flow cytometry and analyzed by ModFit LT 3.0. One representative set of cell cycle patterns of the cell lines upon treatments was presented. **(c)** Percentages of sLCL 381 and **(d)** 421 cells in sub-G1 phases. The results were analyzed for statistical significance using One-way ANOVA Dunnett's Multiple Comparison Test. p-value less than 0.05 was considered statistically significant; $^*$p $< 0.05$, $^{**}$p $< 0.01$, $^{***}$p $< 0.001$, ns = not significant. Error bars

represent the standard error of mean (SEM) of data obtained from three independent experiments.
(TIF)

**S5 Fig. Effects of bortezomib/venetoclax on the growth suppression of sLCL 381 xenografts in SCID mice.** sLCL 381 (1 x $10^7$ cells) were subcutaneously injected into the right flanks of SCID mice, at age of 6–7 weeks. When the mean volume of the tumors reached to approximate 50 mm$^3$, the mice were treated with either 1 mg/kg bortezomib (intraperitoneal (IP) injection in 2% DMSO, 30% PEG (polyethylene glycol) 400 (MedChemExpress, USA) and 69% saline on day 1, 5 and 9, n = 6), 100 mg/kg venetoclax (oral gavage (PO) in 5% DMSO, 60% phosal 50PG (MedChemExpress, USA), 30% PEG400 and 5% ethanol for 5 days per week until day 22, n = 6), bortezomib/venetoclax (n = 6), or DMSO vehicle control (n = 6). **(a)** The mice were euthanized by 150 mg/kg 0.6% pentobarbital via IP injection and **(b)** the tumors were dissected out at the end of experiment (30 days post-treatment). **(c)** Histological analysis confirmed that the tumour cells expressed B-cell marker CD20 by immunohistochemistry, and EBV-encoded small RNA (EBER) by in-situ hybridization (scale bar: 100 μm with 20 μm intervals).
(TIF)

**S6 Fig. Viability of additional sLCLs upon treatment of bortezomib and venetoclax.** (a) sLCL 366, (b) sLCL 397, (c) sLCL 401, and (d) sLCL 478 were treated with 10nM of bortezomib and 5μM of venetoclax for 24 hours and stained by CCK8 kit. OD450nm was measured, and cell viability was plotted. **$p < 0.01$, ***$p < 0.001$, ****$p < 0.0001$.
(TIF)

**S7 Fig. MTT assay and isobologram analysis showing the effect of bort/venetoclax on the cell viability of BL2 EBV-negative/WT EBV pair, BL31 EBV-negative/WT EBV pair, sLCL 381, and sLCL 421.** The cells were treated with combination of bortezomib (0, 1, 2, 4, 8, 16, 32, and 64 nM) and venetoclax (0, 1.25, 2.5, 5, 10, 20 μM) for 24 hours and stained by MTT solution. OD570 nm and OD630 nm were measured, and cell viability was plotted. Percentages of cell viability of (a) BL2 EBV-negative/WT EBV pair, (b) BL31 EBV-negative/WT EBV pair, and (c) sLCL381 and sLCL 421 in bort/venetoclax treatments were determined, and isobolograms were applied for analysis of synergism.
(TIF)

**S8 Fig. Synergy scores computed by SynergyFinder.** The viability of **(a)** sLCL 381 and **(b)** sLCL 421 were loaded to SynergyFinder and the synergy scores were computed.
(TIF)

**S9 Fig. Pathway enrichment analysis of RNA-seq data revealed that bortezomib/venetoclax led to differentially expressed genes in pathways related to EBV infection and cell cycle with top scores.** sLCL 381 and sLCL 421 cells were treated with either DMSO or bortezomib/venetoclax for 24 hours. Total RNA was extracted by using TRIzol (Invitrogen, USA). Preparation of cDNA libraries, Illumina sequencing (NovaSeq 6000) and raw data analysis was performed by Centre for PanorOmic Sciences (CPOS), Genomics Core, LKS Faculty of Medicine, The University of Hong Kong. Differentially expressed genes with FDR < 0.05 were adjusted for multiple testing comparisons using Bonferroni's correction. Pathway enrichment was analyzed by Fisher's Exact test using Partek Genomics Suite 6.6.
(TIF)

**S10 Fig. Pathway enrichment analysis of RNA-seq data illustrating the differentially expressed genes that were induced by bortezomib/venetoclax in pathways related to EBV infection and cell cycle. (a)** sLCL 381 and **(b)** sLCL 421 cells were treated with either DMSO

or bortezomib/venetoclax for 24 hours. Total RNA was extracted by using TRIzol (Invitrogen, USA). Preparation of cDNA libraries, Illumina sequencing (NovaSeq 6000) and raw data analysis was performed by Centre for PanorOmic Sciences (CPOS), Genomics Core, LKS Faculty of Medicine, The University of Hong Kong. Differentially expressed genes with FDR < 0.05 were adjusted for multiple testing comparisons using Bonferroni's correction. Pathway enrichment was analyzed by Fisher's Exact test using Partek Genomics Suite 6.6. The adjusted differentially expressed genes related to EBV proteins and cell cycle were shown.
(TIF)

**S11 Fig. Antibodies used in western blot analysis.**
(TIF)

**S1 Data. Raw data of the charts in Fig 1.**
(XLSX)

**S2 Data. Raw data of the charts in Fig 2.**
(XLSX)

**S3 Data. Raw data of the charts in Fig 3.**
(XLSX)

**S4 Data. Raw data of the charts in Fig 5.**
(XLSX)

**S5 Data. Raw data of the charts in Fig 6.**
(XLSX)

## Acknowledgments

We thank Prof. Jaap Middeldorp (VU University Medical Centre, Amsterdam, The Netherlands) and Prof. Paul Farrell (Imperial College London, UK) for providing the antibodies used in this study. We thank Sam Yuen and DY Jin of The University of Hong Kong for providing the 293T cell line. We are grateful to Mr. Paul Wong, Miss Penny Wong and Mr. Samson Lau of Department of Pathology of The University of Hong Kong for the preparation of the tumor histology slides, immunohistochemical staining and EBER ISH analyses. We acknowledge Prof. Christian Munz of University of Zurich for critically reviewing the manuscript.

## Author Contributions

**Conceptualization:** Kam Pui Tam, Alan K. S. Chiang.

**Data curation:** Kam Pui Tam, Jia Xie, Rex Kwok Him Au-Yeung.

**Formal analysis:** Kam Pui Tam, Jia Xie, Rex Kwok Him Au-Yeung, Alan K. S. Chiang.

**Funding acquisition:** Alan K. S. Chiang.

**Project administration:** Alan K. S. Chiang.

**Supervision:** Alan K. S. Chiang.

**Validation:** Kam Pui Tam, Jia Xie.

**Visualization:** Kam Pui Tam, Jia Xie, Rex Kwok Him Au-Yeung.

**Writing – original draft:** Kam Pui Tam, Jia Xie, Alan K. S. Chiang.

**Writing – review & editing:** Jia Xie, Alan K. S. Chiang.

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
