## [Decision Letter · Decision Letter 0]

10 Jun 2024

Dear Dr Chiang,

Thank you very much for submitting your manuscript "Combination of bortezomib and venetoclax targets the pro-survival function of LMP-1 and EBNA-3C of Epstein-Barr virus in post-transplant lymphoproliferative disorder" for consideration at PLOS Pathogens. As with all papers reviewed by the journal, your manuscript was reviewed by members of the editorial board and by several independent reviewers. In light of the reviews (below this email), we would like to invite the resubmission of a significantly-revised version that takes into account the reviewers' comments.

We cannot make any decision about publication until we have seen the revised manuscript and your response to the reviewers' comments. Your revised manuscript is also likely to be sent to reviewers for further evaluation.

Sincerely,

Benjamin E Gewurz, M.D., Ph.D.

Academic Editor

PLOS Pathogens

Robert Kalejta

Section Editor

PLOS Pathogens

Michael Malim

Editor-in-Chief

PLOS Pathogens

orcid.org/0000-0002-7699-2064

Reviewer's Responses to Questions

**Part I - Summary**

Reviewer #1: Alan Chiang and colleagues report that a combination of bortezomib and venotoclax can eliminate spontaneous LCLs (sLCLs) established from the blood of patients with PTLD and HLH, both in vitro and in vivo. The rationale behind this combination was to simultaneously induce pro-apoptotic signals mediated by Noxa-1 using bortezomib and to block anti-apoptotic signals such as Mcl-1 and Bcl2 with venotoclax. Furthermore, Bortezomib inhibited proteasome-mediated degradation of I-kappa-B-alpha, thereby maintaining NF-kappa-B inactive and sensitizing cells to apoptotic stimuli.

As a result, treated cells showed reduced entry in the cell cycle, increased intrinsic apoptosis and ROS production. Knockdowns of Noxia, BHRF1, EBNA3C and LMP1 showed the importance of these cellular or viral proteins in regulating the LCLs apoptotic status. The authors then used a mouse model to assess the efficacy of the treatment. The combination therapy reduced tumor volume more than 10 times relative to mock controls, and sometimes led to complete regression. There was no evidence of major immediate toxicity associated with the drug treatment.

This is a well-written and concise paper that offers new therapeutic perspectives. There is certainly an unmet medical need for these aggressive tumor types and new treatment options using FDA-approved drugs are more than welcome. The authors have shown the efficacy of the combination therapy and offered mechanistic insights to explain the observed effects. Furthermore, the used knockdowns to link these effects with the expression of viral proteins.

Reviewer #2: Strengths- the authors test a drug combination in vitro and in an animal model which suggests that the combination may be of clinical use. They study the effects of the drug combination in a number of in vitro assays to determine the mechanism of action.

Weaknesses- some of the findings seem overstated and some controls are missing (see major comments)

Reviewer #3: In their manuscript, Tam and colleagues investigated a potential combination therapy for treating Epstein-Barr virus (EBV) associated post-transplant lymphoproliferative disorder (PTLD) in organ transplant patients. The authors demonstrated a novel treatment strategy involving a combination of bortezomib and venetoclax to target key EBV oncoproteins. They showed promising results in killing patient-derived spontaneous lymphoblastoid cell lines. While the results are compelling and generally support the authors' conclusion, there are a few points that need further experimental support.

Specifically:

1) The authors should address drug synergy using a more quantitative analysis method, such as SynergyFinder.

2) The authors mentioned that the combinatorial treatment induces apoptosis by DNA damage triggered by ROS production. However, they did not provide evidence of increasing levels of DNA damage or ROS production to support their claims, as well as the model proposed in Figure 7. They should also confirm DNA damage in tumor sections from mice treated as in Figure 6.

**Part II – Major Issues: Key Experiments Required for Acceptance**

Reviewer #1: 1) What is the evidence that sLCL381 is derived from a PTLD? Under immunosuppression, sLCLs, non-necessarily related to a tumor, are easily established. Thus, the phenotype of the tested sLCLs needs to be better characterized. Does for example this cell line consist of IgM-producing plasmablasts (see doi: 10.1126/scitranslmed.adh8846. Epub 2024 Apr 10.)? Does it carry genetic abnormalities? (See doi: 10.1002/gcc.20287). Are there phenotypic similarities with the clinical tumor they are suspected to derive from? A similar question arises for sLCL421. Does it derive from a histologically proven B cell lymphoma that developed in a patient with HLH? Or is it a non-neoplastic sLCL that grew out in vitro in the context of immune dysregulation and HLH?

2) While it is important to test the effect of venotoclax and bortezomib on sLCLs as the authors did in the paper, I would like to see the effect of this drug combination on a set of LCLs established in vitro. Here it would be sufficient to test the optimal concentrations already identified.

3) Similarly, it would be important to delineate the effect of venotoclax and bortezomib on the establishment of LCLs. Transformation assays using primary B cells and laboratory strains in the presence or absence of the drug combination would be suitable.

Reviewer #2: 1. The study would benefit from testing of the combination in vitro on EBV-negative B cells. These should be B cells purified from human PBMCs (the vast majority of B cells in the blood are EBV-negative so a seropositive donor would be fine) and an EBV-negative human B cell line(s) (e.g. BJAB, DG75, etc.). This is important to know if the drugs are toxic in the absence of EBV infection.

2. The authors use LCLs from pediatric patients with PTLD. Was permission given by an ethics committee and consent/assent obtain to use these human cells from patients for research? I do not see this in the methods.

Reviewer #3: 1) The authors should address drug synergy using a more quantitative analysis method, such as SynergyFinder.

2) The authors mentioned that the combinatorial treatment induces apoptosis by DNA damage triggered by ROS production. However, they did not provide evidence of increasing levels of DNA damage or ROS production to support their claims, as well as the model proposed in Figure 7. They should also confirm DNA damage in tumor sections from mice treated as in Figure 6.

**Part III – Minor Issues: Editorial and Data Presentation Modifications**

Reviewer #1: (No Response)

Reviewer #2: Other

1. The title of the paper and the last sentence of the abstract are misleading. The authors have not tested the drugs in PTLD, only in EBV-transformed B cells from patients with EBV LPD.

2. line 54. It is simplistic to say rituximab is effective for treatment of PTLD. It works in some, but certainly not all cases (as the authors state later in the paper).

3. Line 173 the authors need to indicate what CC3 is.

4. Fig 1D. lines 167-168. The authors state that NAC diminished the activity of bort/venetoclax to reduce CDK4 and cyclin D1. This does not seem to be the case in Fig 1D for sLCL421. The bands for the cell cycle proteins do not increase with NAC.

5. Fig. 2e, Fig.2g. lines 196-198. Knockdown of NOXA does not look like it has much effect on resistance to bortezomib compared to the parental cells (last and first panels in Fig 2e and 2g.

6. Line 202 Is a combination index of 0.4869 with shNOXA really different than that of 0.4489 with that of shLUC. Since the shLUC and parental differ by 0.3 this difference seems like it could just be variation in the assay.

7. Fig 4A, lines 23-224. There is very little difference NF-KB p65 phosphorylation with bort/venetoclax in the either of the shLMP-1 cells. To say there is synergistic reduction is not supported by figure 4A. Similarly lines 225-226 say there is less Bcl-2 and p-Bcl-2(ser) in shLMP1 and shE3C LCLs, but I see little or no difference between bort/venetoclax and any of the other conditions for shE3C sLCL381 cells and for p-Bcl-(Ser) in sLCL381 cells (Fig 4B). I also see no assay for transactivation (line 223). Finally, line 228 refers to Fig 4a and synergy; this figure does not show evidence of synergy.

8. Figure 4C. line 230-231. While the authors say there is stronger CC3 and c-PARP with bort/venetoclax in parental and shLUC than the shBHRF1, shNOXA, shLMP1 and shE3C cells, figure 4C shows strong CC3, but weaker c-PARP.

9. Discussion lines 284-285. The authors show no studies of synergy for DNA damage response or inhibition of DNA repair.

10. Discussion lines 310-322. Are the levels of bortezomib cited in these papers similar to the levels of bortezomib used in the current study; if not these studies may not be relevant to the discussion.

Reviewer #3: (No Response)

PLOS authors have the option to publish the peer review history of their article (what does this mean?). If published, this will include your full peer review and any attached files.

Reviewer #1: No

Reviewer #2: No

Reviewer #3: No
---

## [Decision Letter · Decision Letter 1]

16 Sep 2024

Dear Dr Chiang,

We are pleased to inform you that your manuscript 'Combination of bortezomib and venetoclax targets the pro-survival function of LMP-1 and EBNA-3C of Epstein-Barr virus in spontaneous lymphoblastoid cell lines' has been provisionally accepted for publication in PLOS Pathogens.

Best regards,

Benjamin E Gewurz, M.D., Ph.D.

Academic Editor

PLOS Pathogens

Robert Kalejta

Section Editor

PLOS Pathogens

Michael Malim

Editor-in-Chief

PLOS Pathogens

orcid.org/0000-0002-7699-2064

Please address the minor issues suggested by Reviewer #2 prior to publication. Congratulations

Reviewer Comments (if any, and for reference):

Reviewer's Responses to Questions

**Part I - Summary**

Reviewer #1: The authors have adequately addresed my concerns.

Reviewer #2: The authors have responded to the reviewers comments but a few point remain:

**Part II – Major Issues: Key Experiments Required for Acceptance**

Reviewer #1: (No Response)

Reviewer #2: only minor issues

**Part III – Minor Issues: Editorial and Data Presentation Modifications**

Reviewer #1: (No Response)

Reviewer #2: 1. Part II. Reviewer 2, comment 1. There differences in synergy in Figure S7 between EBV+ and EBV- cell lines, but the differences are not great.

On line 225 I would recommend that the authors wrote “Bort/venetoclax elicited modestly stronger synergistic effects on EBV… “

2. Part III. Reviewer 2. Comment 5. The difference with Noxa knockdown was not large.

I would again qualify the statement on line 202 “Knockdown of Noxa conferred the sLCLs modestly stronger resistance to bortezomib…”

3. Part III. Reviewer 2. Comment 7. Figure 4a. The authors in their rebuttal write ”We agree there is NO difference in p-p65 amongst untreated, single, or combination treatment of shLMP-1 cells (figure 4a).”

Yet line 246 still states “Moreover, the phosphorylation of NF-KBpp65 were synergistically reduced by bort/venetoclax in the LCLs…” I think the authors need to correct line 246 to qualify which cells (shLuc and shE3c) showed reduction.

4. Heading to Figure S6. I would change “extra sLCLs” to “additional “LCLs”

PLOS authors have the option to publish the peer review history of their article (what does this mean?). If published, this will include your full peer review and any attached files.

Reviewer #1: No

Reviewer #2: No

---

## [Editor Report · Acceptance letter]

22 Sep 2024

Dear Dr Chiang,

We are delighted to inform you that your manuscript, "Combination of bortezomib and venetoclax targets the pro-survival function of LMP-1 and EBNA-3C of Epstein-Barr virus in spontaneous lymphoblastoid cell lines," has been formally accepted for publication in PLOS Pathogens.

Best regards,

Michael Malim

Editor-in-Chief

PLOS Pathogens

orcid.org/0000-0002-7699-2064